# Bioengineering the Human Intestinal Mucosa and the Importance of Stromal Support for Pharmacological Evaluation In Vitro

**DOI:** 10.3390/cells13221859

**Published:** 2024-11-08

**Authors:** Matthew Freer, Jim Cooper, Kirsty Goncalves, Stefan Przyborski

**Affiliations:** 1Department of Biosciences, Durham University, Durham DH1 3LE, UK; matthewfreer1011@gmail.com; 2European Collection of Authenticated Cell Cultures, Salisbury SP4 0JG, UK; jim.cooper@ukhsa.gov.uk; 3Reprocell Europe Ltd., Glasgow G20 0XA, UK

**Keywords:** intestine, tissue engineering, drug discovery, fibroblast, keratinocyte growth factor, Caco-2, pharmacology, 3D cell culture, scaffold

## Abstract

Drug discovery is associated with high levels of compound elimination in all stages of development. The current practices for the pharmacokinetic testing of intestinal absorption combine Transwell^®^ inserts with the Caco-2 cell line and are associated with a wide range of limitations. The improvement of pharmacokinetic research relies on the development of more advanced in vitro intestinal constructs that better represent human native tissue and its response to drugs, providing greater predictive accuracy. Here, we present a humanized, bioengineered intestinal construct that recapitulates aspects of intestinal microanatomy. We present improved histotypic characteristics reminiscent of the human intestine, such as a reduction in transepithelial electrical resistance (TEER) and the formation of a robust basement membrane, which are contributed to in-part by a strong stromal foundation. We explore the link between stromal–epithelial crosstalk, paracrine communication, and the role of the keratinocyte growth factor (KGF) as a soluble mediator, underpinning the tissue-specific role of fibroblast subpopulations. Permeability studies adapted to a 96-well format allow for high throughput screening and demonstrate the role of the stromal compartment and tissue architecture on permeability and functionality, which is thought to be one of many factors responsible for unexpected drug outcomes using current approaches for pharmacokinetic testing.

## 1. Introduction

In vitro models of the intestinal epithelium are routinely used in the pharmacological assessment of drugs prior to clinical trials. These models provide a basic screening platform to provide insight into the transport dynamics of target compounds in vivo, including transport receptor affinity through active permeability studies and passive permeability studies through either intra- or inter-cellular diffusion [1,2,3,4]. Widely considered the “gold standard” model for intestinal absorption is the Caco-2 monolayer, cultured on porous polycarbonate inserts [5,6,7,8,9].

In part due to restrictions associated with animal trials, in vitro models provide more attractive predictive tools that act as a pre-clinical screen to streamline the clinical trial process, which is costly, time consuming, and wrought with ethical implications. The Caco-2 monolayer approach has been widely adopted by the pharmaceutical industry as it is quick, inexpensive, and not especially labor intensive [6]. This system provides a simplistic model of the intestinal epithelium that shares some characteristics with the native tissue, including polarized monolayer formation, junctional protein expression, and an apical brush border [10,11,12]. Caco-2 cells were originally isolated from colon adenocarcinoma and their high proliferative capacity lends itself to expanding the cell population, especially for high-throughput screening. However, their simplistic culture in isolation as a monolayer provides a level of variability that may, in part, contribute to the high level of failure of drug compounds when translated from in vitro to in vivo testing [13]. Examples of which, include hydrophilic drugs where low Caco-2 permeability in vitro does not correlate with high absorption of these compounds in vivo such as: amoxicillin, cefadroxil, cephalexin, loracarbef, pregabalin, and zidovudine [14].

Although widely considered by many the “gold standard*”* model of intestinal absorption, the traditional Caco-2 monolayer culture system does not accurately reflect the absorption of a range of molecule types in vivo. For example, the permeability of lipophilic compounds is considered underestimated by the Caco-2 system due to drug retention in the model [15]. Similarly, lipophobic compounds such as atenolol have been reported in the literature to be disparate between the Caco-2 system and in vivo, with an apparent permeability (Papp) of 0.66 reported in Caco-2 monolayers [15,16,17,18,19], compared with 11.5 in rat jejunum and 2.82 in the human intestine [20]. Furthermore, increased expression levels of protein transporters such as MRP-1/2 and BCRP, and decreased expression of MDR1 compared with the native tissue [21,22], has significant effects on carrier-mediated transport [23]. Variability in barrier properties compared with the native tissue also poses a problem when translating in vitro findings in vivo. Caco-2 monolayer cultures have been documented to exhibit abnormal barrier properties compared with in vivo such as high transepithelial resistance (TEER) readings consistent with reduced barrier permeability [24,25], which is not representative of the native tissue, particularly as the primary functional role of the intestine is permeability and absorption (Table 1).

The existing Caco-2 model also suffers from many limitations including morphological and functional variability depending on culture conditions [26,27,28]. Most often Caco-2 models are an extremely simplistic representation of a multi-cellular, complex tissue. The intestine, as with all epithelial tissues, is comprised of multiple layers consisting of specific cell types and extracellular matrix (ECM) constituents that combine to produce a functional tissue [29,30]. The lamina propria is composed of supporting cell types vasculature, and ECM proteins that provide structural, cellular, and biochemical support to the epithelial cells above. In the case of the intestine, the overlying epithelial cells, enterocytes, are a polarized simple columnar epithelium, the surface of which contains a microvilli brush border that is integral to the primary role of absorption [31,32]. Communication between the epithelial and stromal compartments, either through direct contact, structural support, or paracrine communication, is essential for intestinal function in vivo [33]. These factors are an increasingly important consideration for the bioengineering of epithelial tissues in vitro.

Previously, we have demonstrated the importance of the sub-epithelial compartment when constructing a full-thickness skin equivalent, with a particular focus on the role of dermal fibroblasts [34]. Dermal fibroblasts within this tissue construct neosynthesize endogenous ECM to create a foundation upon which a fully differentiated, stratified, and keratinized epidermis forms [35]. We have also demonstrated how communication between the dermal–epidermal compartments through direct cell–cell contact, basement membrane formation, and paracrine mechanisms has a profound impact structurally and functionally on the tissue within the context of skin pigmentation and aging [36,37,38].

In this study, we explore the role of the sub-epithelial compartment within a bioengineered intestinal construct [39], with the aim of improving the current benchmark Caco-2 monolayer approach to intestinal absorption studies in vitro. In the past, we have demonstrated how the choice of fibroblast subpopulation (dermal or intestinal origin) can significantly impact the structure and function of the 3D tissue construct [40]. Here, we explore the possible mechanism of interaction between the cell populations within the engineered tissue. We utilize a variety of methods to investigate the paracrine role of soluble mediators secreted by the fibroblast population and the crosstalk between both epithelial and stromal compartments whilst identifying keratinocyte growth factor (KGF) as a potential mediator. Although paracrine influence was determined to have an impact on tissue functionality, improved intestinal absorption through permeability studies and reduced TEER were the greatest when both submucosal and epithelial compartments were co-cultured together as a single construct, more accurately recapitulating the native anatomy. This suggests that although paracrine crosstalk between compartments is an important factor, geometry and direct cell–cell contact is also a critical consideration when bioengineering epithelial tissue.

## 2. Materials and Methods

### 2.1. Maintenance of Cell Populations

The human colorectal adenocarcinoma cell line, Caco-2 (European Collection of Authenticated Cell Cultures, ECACC, Salisbury, UK) which is commonly used in intestinal barrier models due to its ability to spontaneously differentiate adopting qualities representative of enterocytes, was maintained as per the manufacturer’s instructions. Human fibroblast populations, including neonatal dermal fibroblasts (HDFn, ThermoFisher, Loughborough, UK) and the colonic fibroblast line CCD-18co (ATCC, Glasgow, UK), were maintained up to 80% confluency and were used within 10 passages in 3D models.

All three cell types were maintained in Dulbecco’s Modified Eagles Medium (DMEM, ThermoFisher) supplemented with 10% fetal bovine serum (FBS, ThermoFisher, Waltham, MA, USA), 1 mM sodium pyruvate (ThermoFisher), 2 mM L-glutamine (ThermoFisher), and 1% non-essential amino acids (ThermoFisher). The cells were maintained in standard cell culture flasks at 37 °C, 5% CO_2,_ and 95% humidity.

### 2.2. Epithelial-Only Tissue Construction

Monolayer cultures of Caco-2 cells, representative of the current and heavily used experimental approach (also described as epithelial-only (EO) constructs), were created using Transwell^®^ culture inserts (Corning, Lowell, MA, USA). Caco-2 cells were seeded at a density of 250,000 cells cm^−2^ onto 6-well format Snapwell Transwell^®^ inserts, a porous, polycarbonate substrate of an average pore size: 0.4 μm. EO constructs were cultured for 21–25 days to promote Caco-2 differentiation and polarization, and the medium was changed every 48 h.

For conditioned medium studies, Caco-2 monolayers were established as above, but after 48 h, the medium was replenished with a 1:1 mix of fresh culture medium and conditioned medium obtained from 80% confluent fibroblast cultures (HDFn or CCD-18co). The conditioned medium was centrifuged to remove cellular debris and sterile filtered before addition to EO cultures. Similarly, KGF-supplemented cultures were seeded in standard conditions and supplemented with a range of concentrations (0.5–25 ng·mL^−1^) of KGF following the establishment of cultures for 48 h. Both conditioned and KGF-supplemented media were replenished at each media change: every two days.

### 2.3. Full Thickness Intestinal Tissue Construction

Full thickness (FT) intestinal constructs consisting of both a stromal and an epithelial compartment were constructed using Alvetex^®^ Scaffold (Reprocell Europe Ltd., Glasgow, UK), a porous, polystyrene scaffold membrane with an average pore size of 40 μm. Alvetex^®^ Scaffold was rendered hydrophilic through complete immersion in 70% ethanol prior to washing in phosphate-buffered saline (PBS). Fibroblast populations (HDFn or CCD-18co) were seeded at a density of 250,000 cells cm^−2^ per scaffold and cultured for 7 days. Additional cell seeding steps were required for CCD-18co constructs, with a further 250,000 cells cm^−2^ added at days 7, 9, and 11 of culture. Stromal compartments were maintained for up to 14 days with twice-weekly media changes prior to the construction of the epithelial compartment.

An epithelial compartment was then constructed on the stromal foundation through the addition of Caco-2 cells at a seeding density of 250,000 cells cm^−2^ per tissue model. Tissue constructs were cultured for a further 21–25 days to promote model maturation prior to analysis or downstream testing. Tissue constructs were cultured in a medium consisting of DMEM, 10% FBS, 1 mM sodium pyruvate, 2 mM L-glutamine, and 1% non-essential amino acids for the entire culture duration.

### 2.4. TransEpithelial Electrical Resistance (TEER) Measurement

The transepithelial electrical resistance (TEER) of the EO constructs was measured using an EVOM2 Voltohmeter with STX2 chopstick probes (World Precision Instruments, Hertfordshire, UK). The culture medium was replaced for all cultures 1 h prior to measurement, and measurements were obtained through the immersion of the probes into the culture medium.

The measurement of TEER in 3D constructs was achieved through use of a modified Ussing chamber system (World Precision Instruments). Notably, the chamber module was modified to cater for a 96-well Alvetex^®^ Scaffold insert. Electrodes for electrophysical measurements were created using the tips of 1 mL syringes (voltage electrodes were cut to 2 cm and current electrodes were cut to 3 cm lengths). The 3 mm diameter silver rods were sanded clean and submerged overnight in a sodium hypochlorite solution to create a silver chloride coating. The ends of the silver rods were sanded clean to allow for good connection and connected to the preamplifier equipment. The syringe tips were filled with 100 °C 4% Agar and 3 M KCl solution, and the AgCl rod was inserted whilst the solution was still liquid. The electrode tips were equilibrated to room temperature prior to use. A total of 10 μA was applied across the models for 1 s every 9 s, and voltage deflections were recorded in Labchart 8 software (ADInstruments, Oxfordshire, UK).

### 2.5. Ussing Chamber Drug Permeability Testing

Drug permeability testing was conducted in an Ussing chamber using established methods [20,41,42]. Test compounds (Appendix A) were solubulized in dimethyl sulfoxide (DMSO) (Sigma-Aldrich, Cambridge, UK)) at <0.01% (*v*/*v*) to avoid toxicity. Permeability assays were conducted in a medium consisting of Hank’s Balanced Salt Solution (HBSS, ThermoFisher) supplemented with 2% glucose at 37 °C through use of a circulating water bath. Carbogen gas (95% O_2_, 5% CO_2_, BOC, Guilford, UK) was bubbled through the chamber system to oxygenate the assay medium and provide circulation to the system. Measurements were taken after 120 min and the apparent permeability (Papp) was calculated as follows:$$Papp=\frac{V_{R}dC_{R}}{\begin{array}{c} dt\;A\;C_{D0} \end{array}}$$
$$V_{R}:Volume\;of\;Receiver\;Compartment\;\frac{{dC}_{R}}{dt}:Change\;in\;Analyte\;Concnentration\;of\;Reciever\;Compartment\;with\;Time\;A:Area\;of\;Transport\;Interface\;C_{D0}:Concentration\;of\;Donor\;Compartment\;at\;Time\;Zero$$

### 2.6. Keratinocyte Growth Factor (KGF) ELISA

The medium conditioned by fibroblast populations (CCD-18.co and HDFn) was collected from 14-day matured 2D cultures, having been replaced 24 h prior to collection. A keratinocyte growth factor (KGF) ELISA assay (R&D Systems, Abingdon, UK) was performed on the conditioned media samples, as per the manufacturer’s guidelines. The absorbance was measured using a BioTek™ ELx800 (Winooski, VT, USA) plate reader at 450 nm with a 570 nm correction.

### 2.7. Aminopeptidase Activity Assay

Aminopeptidase activity was quantified through the catalyzation of the cleavage of L-slsninr-4-nitroanilide hydrochloride into the breakdown product 4-nitranline. This was achieved through the addition of 0.5 mL of prewarmed 1.5 mM L-analine-4-nitiroanilide hydrochloride solution (Sigma-Aldrich, St. Louis, MI, USA) to the apical compartment of bioengineered tissue constructs (both EO and FT). The constructs were incubated for 1 h at 37 °C at 100 rpm using an orbital shaker. Following incubation, 100 μL of the cleavage product-containing solution was sampled from each construct and the absorbance was quantified at 405 nm using a BioTek™ ELx800 (Winooski, VT, USA) plate reader. Enzyme activity was expressed as a function of tissue area.

### 2.8. Wax Embedding and Sectioning

Bioengineered intestinal tissue constructs (EO, FT) along with native human intestinal tissue, were processed in the same manner for histological analysis as previously described [34,36,37,43,44]. Briefly, the samples were fixed in 4% paraformaldehyde (Sigma-Aldrich) overnight at 4 °C, prior to dehydration through a series of ethanol baths (30–100%). The samples were then incubated in Histo-Clear (Scientific Laboratory Supplies, Nottingham, UK) alone at room temperature for 15 min, in a 1:1 mix of Histo-Clear and molten wax at 65 °C for 30 min, followed by molten wax alone for 1 h at 65 °C. The samples were then embedded in paraffin wax using plastic molds, sectioned at 5 μm using a microtome (Leica, Wetzlar, Germany), and mounted onto charged microscope slides (ThermoFisher Scientific). The slides were then used for both histological and immunofluorescence processes.

### 2.9. Histological Staining

The samples were deparaffinized in Histo-Clear and then rehydrated from 100% ethanol to deionized water (diH_2_O), prior to being stained with Mayer’s haematoxylin (Sigma-Aldrich) for 5 min. The samples were then washed in diH_2_O and incubated in alkaline alcohol for 30 s to blue the nuclei. The samples were once again dehydrated and counterstained in eosin (Sigma-Aldrich) for 30 s, further dehydrated, and incubated in Histo-Clear prior to mounting with Omni-mount (Scientific Laboratory Supplies). The samples were imaged using a light microscope and the Leica EZ software V4 (Leica).

### 2.10. Immunofluorescence

Wax sections were prepared for staining by de-paraffinization in Histo-Clear followed by rehydration from 100% ethanol to PBS. Antigen retrieval was performed in citrate buffer (pH 6, Sigma-Aldrich) at 95 °C for 20 min. Alternatively, EO constructs were prepared through fixation in a 1:1 ice-cold solution of methanol (Sigma-Aldrich) and acetone (Sigma-Aldrich) for 10 min at −20 °C. From this point onwards whole EO constructs and mounted tissue sections were stained in the same manner.

The samples were blocked for 1 h at room temperature in a solution consisting of 10% normal goat serum (NCS, Sigma-Aldrich) and 0.1% Triton X-11 (Sigma-Aldrich) in PBS. The relevant primary antibody (Appendix A) was diluted in PBS and incubated with the samples for 1 h at room temperature, followed by three 5 min washes in PBS. The samples were then incubated with the relevant fluorescent-tagged secondary antibody (Appendix A) and counterstained with Hoechst 3342 (ThermoFisher Scientific) for 1 h at room temperature. Finally, the samples were washed three times for 5 min in PBS and mounted using the Vectashield anti-fade mounting medium (Vector Laboratories, Peterborough, UK). The samples were imaged using the Zeiss 880 confocal microscope with Zen Software (Carl Zeiss, Cambridge, UK, V 13.0).

### 2.11. qPCR Detection of Gene Expression

EO tissue constructs were lysed in RLT buffer (Qiagen, Manchester, UK) and homogenized through repeated aspiration with a 20-gauge needle. The mRNA was extracted and purified through the use of the RNeasy mini kit (Qiagen) and the manufacturer’s instructions were followed. The mRNA was suspended in molecular-grade water and stored at −80 °C prior to analysis. mRNA abundance was quantified using a Nanodrop Spectrophotometer ND-100™ and 200 ng was reverse-transcribed using the GoScript™ Reverse Transcriptase kit (Promega, Hampshire, UK), as per the manufacturer’s instructions. Previously published primer sequences (Appendix A [45,46,47] ) specific to target genes were used, along with SYBR^®^ Green (Bio-Rad, Hertfordshire, UK), and the manufacturer’s instructions were followed. Data were analyzed through normalization to the housekeeping reference genes (*GAPDH*) using the ΔΔCt method [48] and the fold change was averaged after this normalization.

### 2.12. Statistical Analysis

Data sets were analyzed in the GraphPad Prism software (latest version v 10.0.3.). The statistical significance was determined using a Student’s *t*-test or one-way/two-way ANOVA with Tukey’s post hoc test, as appropriate. * = *p* ≤ 0.05, ** = *p* ≤ 0.01, *** = *p* ≤ 0.001, **** = *p* ≤ 0.0001

### 2.13. Human Tissue Ethical Statement

Human small intestine tissue was collected by Biopta (Glasgow, UK) under appropriate ethical protocols and consent, in compliance with local laws and regulations.

## 3. Results

### 3.1. The Role of Tissue-Specific Fibroblast Subpopulations in the Development of a Full-Thickness Bioengineered Intestinal Constructs

Full-thickness human intestinal tissue constructs consisting of both a stromal and an epithelial compartment were bioengineered, as previously described [48]. To better understand the role of tissue-specific fibroblast populations in bioengineering, FT constructs were built upon stromal foundations populated with either intestinal fibroblasts (CCD-18co) or dermal fibroblasts (HDFn). The fibroblasts were allowed to populate the supporting scaffold and neosynthesize endogenous human extracellular matrix (ECM) components to support the overlying epithelial compartment. In order to prevent epithelial cell infiltration, intestinal fibroblast compartments required additional cell seeding steps at days 7, 9, and 11 of culture. Once the stromal compartments reached maturity (21*–*25 days), Caco-2 cells were seeded onto each submucosal foundation and allowed to develop over a 25-day culture window (Figure 1A).

Histological analysis reveals differences in epithelial morphology between FT constructs built upon an intestinal fibroblast (Figure 1B(i)) or a dermal fibroblast (Figure 1B(ii)) stromal foundation. Caco-2 cells seeded upon an intestinal fibroblast-populated compartment generally appeared more columnar and polarized than those seeded upon a dermal fibroblast-populated compartment, and therefore appear more representative of the native human intestine (Figure 1B(iii)).

An essential function of the fibroblast population within these constructs is to neosynthesize endogenous ECM components, thereby providing structural and biochemical support to the tissue. This is advantageous over other widespread approaches as the cells produce these proteins endogenously, without the addition of animal-derived coatings or matrices, leading to a more physiologically representative tissue construct. Stromal compartments populated by intestinal fibroblasts were found, via immunofluorescence analysis, to be rich in collagen I (Figure 1B(iv)), collagen III (Figure 1B(vii)), and collagen IV (Figure 1B(x)), which was comparable to human intestinal tissue (Figure 1B(vi,ix,xv)). However, immunofluorescence staining of stromal compartments populated by dermal fibroblasts was found to be less intense for collagen I (Figure 1B(v)) and collagen III (Figure 1B(viii)), and comparable for collagen IV (Figure 1B(xi)). This suggests that the abundance of collagens produced by the specific fibroblast subpopulations is varied and that the ECM composition secreted by the intestinal fibroblast population may better represent that of the native intestinal tissue.

E-cadherin, a cell–cell junctional component widely expressed in epithelial tissues and associated with epithelial polarization [49], is detectable in the epithelial layer of both bioengineered tissues (Figure 1B(xiii,xiv)) and the native intestinal tissue (Figure 1B(xv)). However, the efflux transporter protein MDR1, typically expressed on the apical membrane of enterocytes [50] appears to be intensely stained on the basolateral surface of epithelial cells when co-cultured on a foundation consisting of dermal fibroblasts (Figure 1B(xvii)). This is in contrast to bioengineered constructs containing intestinal fibroblasts (Figure 1B(xvi)) or native intestinal tissue (Figure 1B(xviii)).

In addition to the structural and biochemical variation between the tissues constructed utilizing different fibroblast subpopulations, functional differences were also identified in terms of barrier properties. Transepithelial electrical resistance (TEER, Figure 1C) was significantly higher in FT constructs built upon a dermal fibroblast stromal foundation. This signifies increased barrier function and, therefore, reduced permeability, which is a known limitation of existing Caco-2 2D monolayer cultures, a difference that less than accurately reflects that of the native tissue.

### 3.2. Paracrine Crosstalk Between Stromal and Epithelial Compartments Significantly Impacts Tissue Structure and Function

In order to better understand and probe the molecular relationship between the epithelial and stromal compartments in the FT tissue constructs, we performed further in vitro testing to identify the role of paracrine crosstalk between the compartments. This was of particular importance as epithelial tissues are known to be impacted by the secretion of soluble mediators from fibroblast populations, basement membrane formation between the compartments, direct cell–cell contact, and biochemical signals transduced via ECM constituents. 

To elucidate the role of paracrine communication between fibroblast and epithelial cell populations, a conditioned medium from both dermal and intestinal fibroblast cultures was collected (Figure 2A). Caco-2 monolayers (epithelial-only construct, EO) were cultured on permeable polycarbonate inserts, a model system that is widely described in the literature [5,10,12]. However, to determine the role of fibroblast-secreted mediators, the conditioned medium was applied to EO constructs for the duration of the epithelial maturation stage of culture.

Histological evaluation of EO constructs cultured in the standard culture medium (Figure 2B(i)) reveals epithelial cells with a flattened morphology reminiscent of a simple squamous epithelia with little evidence of polarization. In contrast, those cultured in intestinal fibroblast conditioned medium (Figure 2B(ii)) appear taller, with basally located nuclei suggestive of polarization and the presence of microvilli at high magnification. However, when EO constructs were cultured with dermal fibroblast conditioned medium (Figure 2B(iii)), their morphology appeared drastically different compared with controls or those cultured in intestinal fibroblast-conditioned medium. Ccell height appeared substantially increased, with cell layering evident across the length of the sample. These stark morphological differences observed across culture conditions demonstrate the powerful capability of paracrine mediators to influence cellular structure and function.

At a molecular level, gene expression analysis pertaining to proteins such as MDR1 (an efflux transporter and mediator of intestinal permeability), villin (an actin-binding protein present in the brush border), and occludin (a tight junction component) also differed significantly across culture conditions. Similarly, the expression of MDR1 (Figure 2C) and villin (Figure 2D) was significantly increased in EO constructs cultured in the intestinal fibroblast-conditioned medium. Whilst expression of occludin (Figure 2E), was decreased in both the intestinal and dermal fibroblast conditioned media conditions. These data suggest that both transporter-mediated and junctional permeability is increased when EO are cultured in the presence of soluble factors secreted by intestinal fibroblasts, as is Caco-2 differentiation and the ability to form microvilli representative of the brush border.

Functionally, TEER (Figure 2F) was substantially reduced in EO constructs cultured with both the intestinal and dermal fibroblast conditioned medium and increased with time in culture across all conditions. This observation supports that of the gene expression analysis, suggestive of cell and molecular level differences in the epithelial barrier when EO constructs are cultured in the presence of fibroblast-secreted soluble components.

In order to identify a potential candidate mediator present in the conditioned medium that could be, in part, responsible for the morphological, gene expression, and functional changes observed in the EO constructs, keratinocyte growth factor (KGF) was considered. We speculated that KGF may play a role in the paracrine communication between the cell populations as it is secreted by fibroblast populations in vivo [49], exerts its effects on epithelial cells [49]*,* and is known to have a homeostatic and protective role in intestinal maintenance [51,52,53], particularly in terms of barrier function and tight junction expression. The level of KGF (Figure 2G) within medium conditioned by intestinal fibroblasts was significantly increased compared with dermal fibroblast conditioned medium and may be partly responsible for the differential response of EO constructs cultured in each condition.

Aminopeptidase activity (Figure 2H), a digestive enzyme secreted by epithelial cells of the human intestine, was also compared between EO constructs cultured in intestinal fibroblast conditioned medium and FT cultures, whereby Caco-2 and intestinal fibroblasts were co-cultured within the same construct. These data demonstrated that although aminopeptidase activity was significantly increased both in paracrine (EO conditioned medium) and direct contact (FT co-culture) models compared with EO controls, there was no significant difference between both fibroblast conditions. These data further support the substantial contribution of paracrine factors in mediating communication between the tissue compartments.

To further explore the role of the KGF as a soluble mediator, capable of cellular influence over the epithelial compartment, exogenous addition of KGF to EO constructs was conducted. Caco-2 cells were seeded upon permeable membranes and were allowed to adhere for 24 h prior to the addition of a range of KGF concentrations (0.5*–*25 ng.mL*^−^*^1^) for the duration of the maturation period (Figure 3A).

Histological analysis revealed flattened epithelial cells reminiscent of a simple squamous epithelium in the absence of KGF supplementation (Figure 3B(i)). However, with increasing KGF supplementation (Figure 3B(ii–iv)), the cells appear progressively more columnar, with their morphology appearing polarized due to basolaterally located nuclei and visible microvilli at 25 ng.mL^−1^ KGF supplementation.

Immunofluorescence staining of junctional proteins E-cadherin (Figure 3B(v–vii)) and occludin (Figure 3B(ix–xii)) is consistent across both control EO constructs and those supplemented with KGF, with staining located at the periphery of the cells, as expected. However, control EO constructs appear heterogeneous, with large irregularly shaped cells across the construct (white arrows), whereas those treated with KGF appear more homogeneous and tightly packed in the epithelial layer. The immunofluorescent detection of villin, a structural component of the microvilli of the brush border that is suggestive of cellular polarization and differentiation, increases with KGF concentration (Figure 3B(xiii–xvi)). Untreated control EO constructs express low levels of villin, apparently isolated to the periphery of cells; however, expression increases with KGF concentration up to 5*–*25 ng.mL*^−^*^1^, with expression ubiquitous and intense across the whole epithelial layer.

Functional measurements such as TEER and aminopeptidase activity also correlate with increasing KGF supplementation. TEER, a functional measurement of epithelial barrier integrity, decreased with increasing KGF concentration (Figure 3C), whilst aminopeptidase activity, a digestive enzyme key to the function digestive function of the intestine in vivo, increased with KGF concentration (Figure 3D).

At the molecular level, the expression of the gene responsible for villin transcription increased with KGF concentration (Figure 3E), consistent with immunofluorescence analysis and supporting the notion that epithelial layers have become more polarized and differentiated, and the presence of microvilli increased. Similarly, expression of the gene responsible for occludin expression (Figure 3F) supports the functional assessment of TEER, in that it decreased with increasing KGF concentration, and may be one of a number of factors that contribute to decreased barrier function.

The detailed analysis undertaken in regard to EO constructs treated with a range of concentrations of KGF displays similar trends to those supplemented with intestinal fibroblast-conditioned medium (Figure 2). These include an increase in villin expression, a decrease in occludin expression, decreased TEER, increased aminopeptidase activity, and morphological changes. Together, these data support the concept that KGF may be a potential paracrine mediator involved in the crosstalk between stromal and epithelial compartments within this bioengineered system.

### 3.3. Permeability of Bioengineered Constructs Is Influenced by Both Paracrine Cellular Communication and Direct Contact Between Compartments

The Ussing chamber system is commonly used to assess intestinal permeability in tissue explants [54].

In this study, we described the adaptation of FT intestinal constructs into a 96-well format Alvetex*^®^* insert (Figure 4A) that was subsequently placed within the chamber of a modified Ussings system, which was, in part, constructed in-house. Either 96-well format FT or EO intestinal constructs were mounted between the chambers of the Ussing system, fed by two reservoirs, and encircled by a circulating water bath that maintained a constant temperature of 37 °C (Figure 4C). Carbogen gas was bubbled through the reservoirs both to oxygenate and circulate the system, while tissue constructs acted as a singular interface between the two reservoirs. This allowed for the assessment of basal–apical (B-A) transport or apical–basal (A-B) transport dependent upon the reservoir with a test analyte that was added or sampled (Figure 4B). This adaptation of the Ussing chamber system then allowed for permeability and physiological measurements to be obtained from bioengineered human intestinal equivalents (Figure 4D).

Rhodamine 123 was selected as a compound to elucidate the extent of MDR1-mediated transport in either the conditioned medium EO constructs or the FT constructs built upon an intestinal or dermal fibroblast stromal foundation. Rhodamine 123 was transported in the apical–basal direction through paracellular transport whereby the compound passed through intercellular spaces between the epithelial cells and was controlled by tight junction expression (Figure 5A). However, in the basal–apical direction, Rhodamine 123 transport is mediated by the efflux transporter, MDR1, which in turn is inhibited by verapamil (Figure 5B).

In the apical-basal direction, EO constructs cultured in the medium conditioned by both intestinal and dermal fibroblasts displayed increased apparent permeability (Papp) to Rhodamine 123, which, as expected, was unaffected by the addition of verapamil (Figure 5C). However, in the basal-apical direction of transport, Papp was enhanced in EO constructs cultured in the dermal fibroblast conditioned medium and was significantly reduced in the presence of verapamil (Figure 5D). This is suggestive of increased MDR1 activity in the basal-apical direction.

However, the transport dynamics displayed by FT constructs built upon intestinal or dermal fibroblast foundations differed from that of EO-conditioned medium studies. In the apical–basal direction, there was no significant difference in Papp between the FT constructs regardless of verapamil treatment, as expected (Figure 5E). Yet, in the basal–apical direction, Papp was significantly increased in the absence of verapamil in FT constructs built upon an intestinal fibroblast stromal foundation, compared with their dermal fibroblast counterpart (Figure 5F). Similarly, in the presence of verapamil, Papp was significantly reduced in those FT constructs that include intestinal fibroblasts and remained relatively unchanged in those inclusive of dermal fibroblasts. This is suggestive of a greater level of MDR1-mediated transport in those FT constructs built upon a foundation of intestinal fibroblasts, conversely to the results obtained from conditioned medium studies.

The supplementation of EO constructs with KGF increased the basal-apical transport of Rhodamine 123 in the absence of verapamil, which was almost completely inhibited by the presence of verapamil (Figure 5G). Together, these data suggest that the KGF is a potential soluble mediator, responsible for the paracrine communication between the cell types and capable of cellular changes consistent with increased permeability. However, the discrepancy between the conditioned medium EO system and the FT constructs suggests that the overall function of the bioengineered tissue is multifactorial and dependent upon both chemical and physical interactions.

Lucifer yellow, a fluorescent molecule that is transported passively through the intercellular space in a paracellular manner, a process regulated by tight junctions, was also used to assess the permeability of the bioengineered tissue constructs (Figure 6A). Although a slight increase in basal–apical transport of EO constructs cultured in the intestinal fibroblast-conditioned medium was detected, it was not significant (Figure 6B). However, in both the apical–basal and basal–apical directions, FT constructs built upon a stromal foundation of intestinal and dermal fibroblasts displayed significantly increased permeability compared with the Caco-2 monolayer EO approach, and thus providing further evidence that both physical and chemical communication between different cell types can enhance the physiological relevance of the system.

The chemotherapeutic drug, methotrexate, is transported in the apical-basal direction by RFC1 and PCFT (Figure 6C); therefore, Papp in this direction is indicative of their function. Apical-basal transport was increased only in FT co-culture constructs and to a greater degree in those co-cultured with dermal fibroblasts (Figure 6D), and therefore suggestive of greater RFC1 and PCFT function in those constructs. Similarly, BCRP-mediated transport in the basal-apical direction, although modestly enhanced in EOs cultured in the intestinal fibroblast conditioned medium, was increased to the greatest extent in FT constructs consisting of a dermal fibroblast-based stromal foundation.

The transport of etoposide, an anti-tumor drug, is mediated by a number of transporter proteins in the basal–apical direction: MDR1, MRP2, BCRP, and OCT1, whereas in the apical–basal direction, is reliant upon paracellular transport (Figure 6E). Similarly to the other compounds tested, a modest increase in paracellular transport was determined in EO constructs cultured in the fibroblast conditioned medium, detected by an increase in Papp in the apical-basal direction (Figure 6F). However, receptor-mediated transport in the basal-apical direction was increased to the greatest degree in FT constructs co-cultured with dermal fibroblasts, further suggestive of a substantial role of direct physical contact between the two cell populations in the regulation of permeability.

Collectively, these data support the notion that although paracrine communication is an important determinant of functionality, direct physical interactions between cells also contribute extensively to the structural and functional phenotype of the tissue.

## 4. Discussion

The current popular approach for intestinal permeability testing in vitro utilizes the Caco-2 monolayer model, which is routinely employed across both industrial and academic sectors [21,55,56,57,58,59]. Although robust, the simplistic monolayer approach to the Caco-2 culture has limitations that reduce its predictive accuracy, such as artificially high TEER and poor paracellular permeability [24,25], both of which reduce the absorptive capacity of the construct and therefore its physiological relevance compared with the native tissue. This study builds on our previous findings [40,60] and describes the development of an intestinal construct containing both stromal and epithelial compartments that better recapitulate aspects of tissue functionality in comparison to the native tissue. This includes structural hallmarks of improved epithelial morphology, increased permeability, cellular transporter expression, and improved functionality, although the expression of metabolic enzymes was outside the scope of this study expression, which is known to be a disparate expression in Caco2 monolayer cultures compared with in vivo cultures. To better understand the underlying cellular and molecular signaling events that give rise to this enhanced phenotype within the system, we also examined the role of the stromal compartment in tissue homeostasis and paracrine communication.

Stromal influence upon the epithelia begins during embryonic development and continues throughout life [61]. Stroma are thought to elicit effects on the epithelia through the secretion of diffusible factors [62], biochemical interactions with the ECM [63], and direct cell–cell contact [63], having the greatest impact on tissues considered to have a rapid cellular turnover such as the intestine, skin, and the female reproductive system [61]. Previously, we have demonstrated the important role of dermal fibroblasts in skin pigmentation [36] and dermal–epidermal ageing crosstalk [64]. Similarly, we have also developed an intestinal construct consisting of both stromal and epithelial compartments, with characteristics reminiscent of the native tissue [40]. However, in this study, we explored the role of stromal–epithelium interactions within the context of intestinal tissue engineering to provide fundamental insights into the cellular mechanisms that govern tissue homeostasis which can lead to improvements in current practices for in vitro permeability testing.

The bioengineered human tissues produced in our laboratory were based on the concept that fibroblasts seeded within the scaffold synthesize tissue-specific endogenous ECM components and signals. This eliminates the use of animal derivatives that are ubiquitous amongst other published intestinal constructs [65,66,67,68], and tackles issues associated with batch–batch variability, poor standardization, and species specificity. The main constituents of the intestinal ECM are collagens I, III, IV, V, VI, and VII, laminin, and fibronectin, alongside glycosaminoglycans such as heparin and chondroitin sulfate, which represent a complex network of proteins that provide biochemical and structural support to the overlying epithelium [69]. In our bioengineered intestinal construct, we demonstrated the varied ECM produced endogenously by both intestinal and dermal fibroblast populations, and that when produced by the intestinal fibroblast population appeared more representative of the native tissue, rich in collagens. ECM composition is particularly important as biochemical cues transduced through cell–matrix interactions have been reported to impact epithelial differentiation [70]. The varied composition produced by the two fibroblast subpopulations discussed herein could therefore be one potential mechanism responsible for the differential functionality of the constructs.

However, ECM interactions are but one of many proposed mechanisms that govern stromal–epithelium communication. Therefore, a reductionist approach was used to adapt the existing epithelial-only (EO) model in order to explore the role of paracrine mediators involved in crosstalk between the compartments. The conditioned mediums from both intestinal and dermal fibroblast populations, when combined with the EO system, resulted in differential cellular responses at both a structural and functional level. The dermal fibroblast-conditioned medium resulted in significant morphological changes and the multilayering of Caco-2 cells. In contrast, the intestinal fibroblast-conditioned medium resulted in the increased expression of the transporter protein MDR1, decreased junctional protein expression (occludin), decreased TEER, and increased villin expression, all of which are consistent with improved functionality and better represent the physiological state of the native intestine [71,72,73].

KGF was considered a potential candidate soluble factor responsible for the paracrine action of intestinal fibroblasts upon the Caco-2 epithelium particularly as KGF is documented to play an important role in intestinal maintenance and function, along with being described as a potent mitogen in many different epithelial systems, responsible for regulation, differentiation, and cellular protection [74]. Specific to the intestine, KGF is known to be a pivotal factor in the maintenance of barrier integrity, epithelial homeostasis, protection, and repair [75,76,77]. Our data showed that intestinal fibroblasts secreted significantly more KGF than their dermal counterparts. This is consistent with accounts across the literature which have previously demonstrated the secretion of high levels of KGF from the intestinal fibroblast line described herein (CCD-18co). The data described in this study support that concentration dependent changes in the EO phenotype were consistent with improved physiological relevance of the construct, such as increased villin expression, decreased TEER, decreased occludin expression, increased aminopeptidase activity, and increased MDR1-mediated transport. These observations are associated with increased permeability and hold particular relevance to the development of improved in vitro systems capable of modeling in vivo intestinal permeability.

KGF is one of many soluble mediators capable of paracrine communication between stromal and epithelial compartments, but many factors contribute to the crosstalk responsible for tissue homeostasis. There is evidence across the literature to implicate other signaling components such as FGF-10, EGF, and IL-6, which have been demonstrated to be secreted by fibroblasts and have influence over Caco-2 function, such as increasing MDR1-mediated transport [51,78,79]. However, paracrine communication is one of multiple mechanisms of interaction between the stromal and epithelial compartments; the data described in this study support the notion that several contributing factors influence the overall phenotype of the tissue construct and should be considered when bioengineering any epithelial tissue.

Our data showed that the presence of both a Caco-2 epithelium with support from a stromal compartment in direct contact resulted in the most functionally relevant tissue construct within the context of permeability testing. This is most likely a combination of endogenous ECM secretion, paracrine communication, and the formation of a basement membrane de novo separating the epithelial from the stromal compartment. The basement membrane is integral in maintaining barrier resistance, cellular attachment, and differentiation, the impairment of which has been demonstrated to lead to cellular dysregulation [79].

The data described herein support the notion that the bioengineering of epithelial tissue in vitro can be complex, with many considerations required to produce a model whose structure and function are reminiscent of the native tissue (summarized in Table 1). These include the selection of an appropriate fibroblast subpopulation (as demonstrated here, distinct populations of fibroblasts express varied properties and differential secretomes), the influence of soluble factors, direct cell-cell contact, basement membrane formation, and ECM composition. We also described the development of a Caco-2-based intestinal construct built upon a foundation of intestinal fibroblasts that demonstrates enhanced biomarker expression, improved cellular morphology, and increased permeability, therefore better recapitulating aspects of intestinal physiology compared with current Caco-2 models. We also demonstrated its assessment in a bespoke Ussing chamber, consistent with industrial standards, capable of high throughput drug, compound, and formulation screening, including a 96-well insert format. The combination of such technologies may have a significant impact on both industrial and academic applications, both providing a pre-clinical predictive tool and a platform capable of providing fundamental insights relevant to intestinal physiology.

## 5. Conclusions

This study provides valuable insight into the complex interactions between stromal and epithelial compartments within a tissue. Many factors contribute toward the overall phenotypic activity of the tissue and, therefore, provide significant challenges when replicating that tissue in vitro. Through a combination of tissue-specific fibroblasts and a 3D geometry allowing cell-cell and cell-matrix interactions to form along a de novo basement membrane, we have demonstrated an FT intestinal construct that better recapitulates intestinal permeability compared with the existing Caco-2 mono-culture approach. Not only does this construct better represent enhanced permeability comparable to a native tissue but is also amenable to high throughput screening of drugs or other compounds.

**Table 1 cells-13-01859-t001:** A comparison of tissue engineering approaches adopted to model aspects of the intestinal structure and function. An overview of current approaches reported in the literature which are used to model intestinal barrier function and permeability. Model systems are compared to the widely accepted Caco2 monolayer approach in terms of extracellular matrix (ECM) inclusion, transepithelial electrical resistance (TEER) as a measure of barrier integrity, notable protein expression, and permeability to small molecules. The approaches utilized in this study are also detailed for comparison including the addition of intestinal fibroblast conditioned medium, keratinocyte growth factor (KGF) supplementation, and direct co-culture of epithelial and stromal cells within a 3D environment (full thickness construct). Arrows represent relative expression compared with standard Caco2 monolayer.

Model System	Compartments Modelled	Cell Lines	ECM	TEER	Protein Expression	Permeability	Citations
Caco2 Monolayer	Epithelial	Caco2	None	+++	Variable	+	[80,81,82,83]
Caco2/HT29-MTX Monolayer *Inclusion of Mucus Producing Cells*	Epithelial	Caco2 and HT29-MTX	None	+	↓ PGP	↓ With co-culture	[14]
Organoids	Epithelial and Stromal	Differentiated from Pluripotent Stem Cells or Isolated from Biopsy	Exogenous Hydrogel	-	Variable—beneficial for disease modeling	Unable to determine directional permeability	[84,85]
Hydrogel-based Construct Featuring Villi Structure	Epithelial and Stromal	Caco2	Exogenous Hydrogel	+	Recapitulation of Villus Structure	++	[86]
Decellularized Matrix	Epithelial and Stromal	Primary Intestinal Crypt Organoids	Porcine or Murine Decullarized Matrix	+	Keratin 18 Villin Muc1/2 Chromogranin	+	[87,88]
Electrospun Scaffold	Epithelial and Stromal	Caco2	Polylactic Acid (PLA) or Poly(ε-caprolactone) (PCL) Nanofibre Scaffold	++	↑ PGP ↑ALP	+	[89,90]
Bioprinted Tissue	Epithelial and Stromal	Primary Human Enterocytes and Primary Human Intestinal Myofibroblasts	Hydrogel Bioink	+	↑ Keratin 19 ↑ PGP ↑ BCRP	+	[91]
Caco2 Monolayer + *Intestinal Fibroblast Conditioned Medium*	Epithelial and Stromal Paracrine Support	Caco2	None	++	↑ MDR1 ↑ Villin ↓ Occludin ↑ Aminopeptidase	++	Described in this study
Caco2 Monolayer + *KGF Supplementation*	Epithelial	Caco2	None	++	↑ Villin ↓ Occludin ↑ Aminopeptidase	++	
3D Full Thickness Construct	Epithelial and Stromal	Caco2 and CCD-19co	Endogenous	+	Collagens I, III, IV E-cadherin MDR1	+++	

## Figures and Tables

**Figure 1 cells-13-01859-f001:**
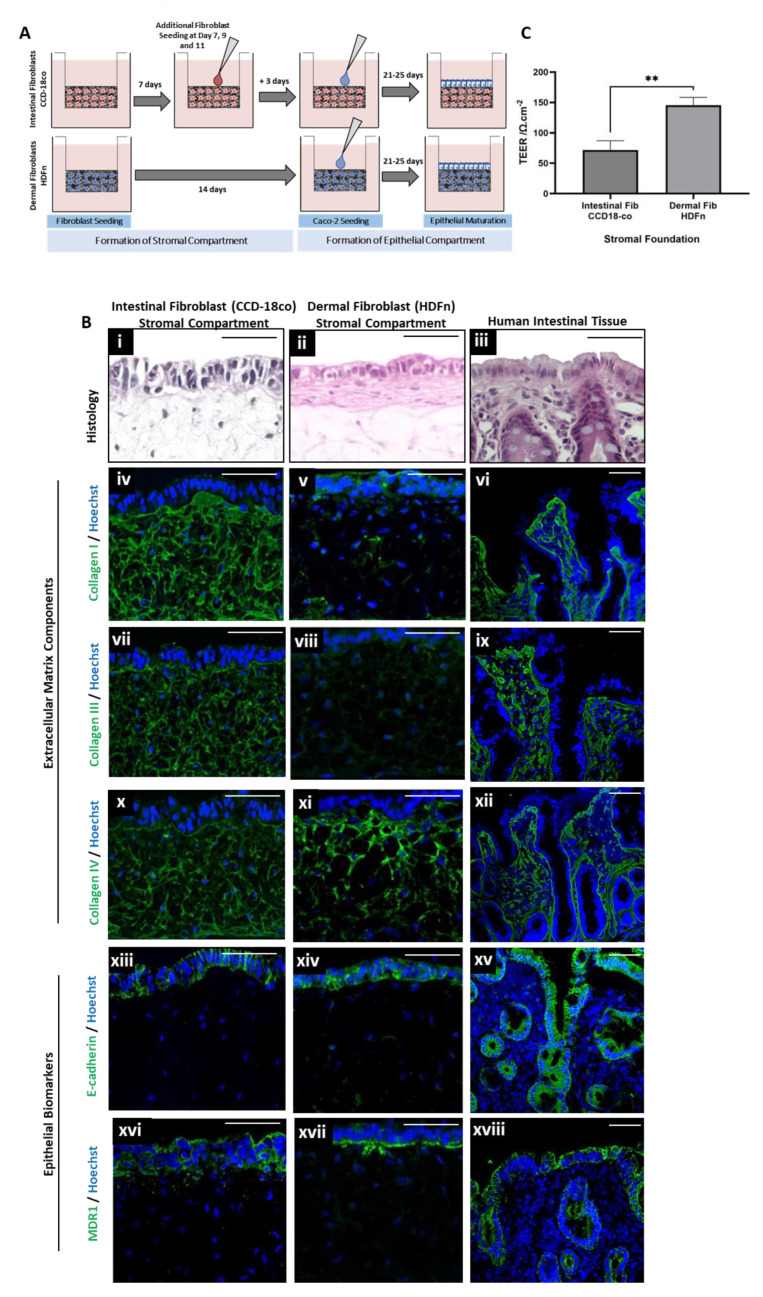
The incorporation of tissue-specific fibroblast subpopulations into the stromal compartment significantly impacts intestinal construct structure and function. (**A**) A schematic representation of full thickness (FT) intestinal constructs bioengineered upon stromal compartments populated by either intestinal (CCD-18co) or dermal (HDFn) fibroblast subpopulations. (**B**i–iii) H&E staining depicts the tissue morphology of bioengineered constructs and native human tissue, both of which consist of a stromal foundation, upon which an epithelial layer resides. Immunofluorescence analysis of extracellular matrix components such as (**B**iv–vi) collagen I, (**B**vii–ix) collagen III, and (**B**x–xii) collagen IV reveal significant expression in the stromal compartment (green) with nuclei stained in blue by Hoechst. Immunofluorescence analysis of junctional protein (Bxiii–xv) E-cadherin, (green, nuclei stained in blue) is specific to the epithelial compartment in all cases, as is the expression of the transporter protein (**B**xvi–xviii) MDR1 (green, nuclei stained in blue). (**C**) Transepithelial electrical resistance (TEER, data represent mean ± SEM, n = 4) is significantly increased when tissue constructs contain a HDFn populated stromal foundation. Scale bars: 50 μm. ** = *p* ≤ 0.01.

**Figure 2 cells-13-01859-f002:**
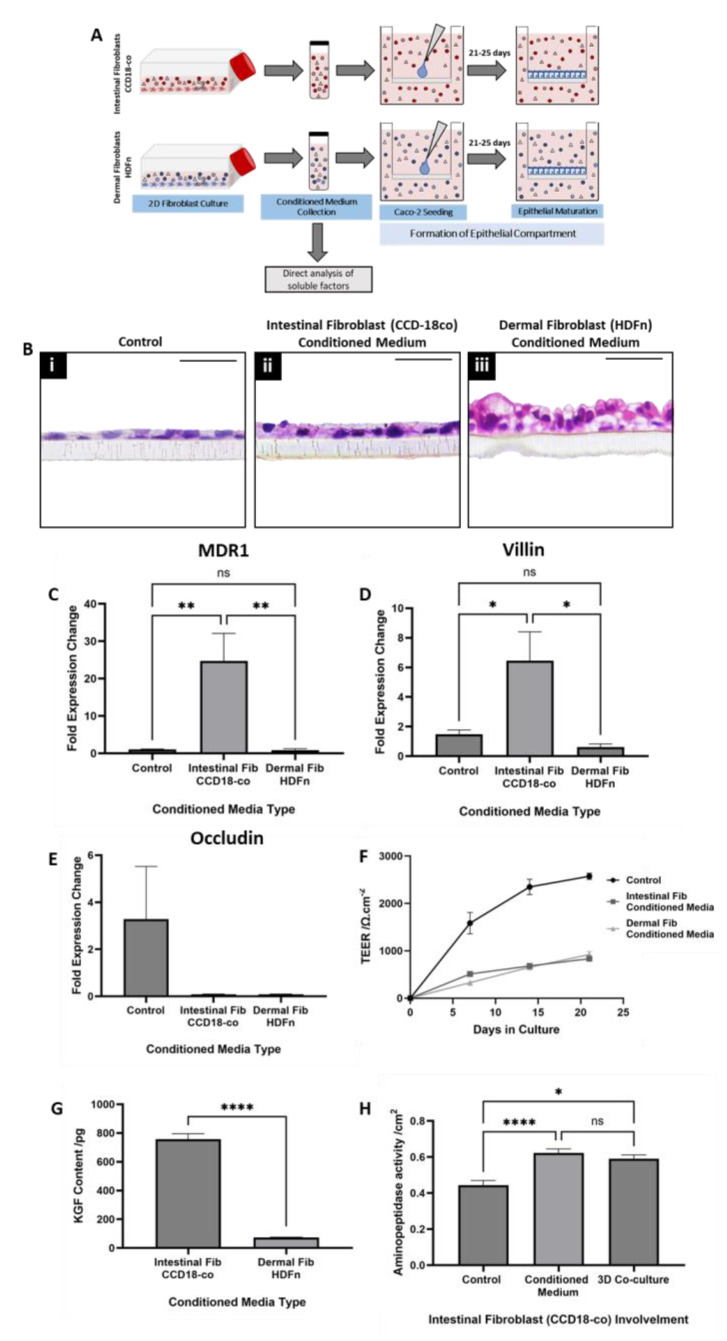
Paracrine factors secreted by fibroblast subpopulations impact intestinal epithelial structure and function in vitro. (**A**) A schematic representation of fibroblast-conditioned medium collection and addition to epithelial-only (EO) Caco-2 monolayer cultures for the duration of the 25-day culture period. Representative H&E staining of EO cultures grown in (**B**i) a standard culture medium or a medium (**B**ii) conditioned by intestinal fibroblast (CCD18-co) or (**B**iii) dermal fibroblast (HDFn) populations. qPCR gene expression analysis of EO constructs cultured in control or fibroblast-conditioned medium, representing genes that encode the transporter protein (**C**) MDR1, (**D**) brush border associated protein villin, and (**E**) tight junction protein occludin (data represent the mean ± SEM, n = 3). (**F**) Transepithelial electrical resistance (TEER) measurements reveal reduced barrier function and, therefore, increased permeability across the duration of the culture period for constructs cultured in the fibroblast conditioned medium (data represent the mean ± SEM, n = 4*–*12). (**G**) ELISA determination of the keratinocyte growth factor (KGF) content of the conditioned medium collected from both intestinal and dermal fibroblasts demonstrates a significant difference between the populations (data represent the mean ± SEM, n = 3). (**H**) Aminopeptidase activity is significantly increased both in EO constructs cultured in the intestinal fibroblast-conditioned medium and full thickness (FT) constructs containing both Caco-2 and intestinal fibroblasts (data represent the mean ± SEM, n = 4*–*15). Scale bars: 50 μm. * = *p* ≤ 0.1,** = *p* ≤ 0.01, **** = *p* ≤ 0.0001.

**Figure 3 cells-13-01859-f003:**
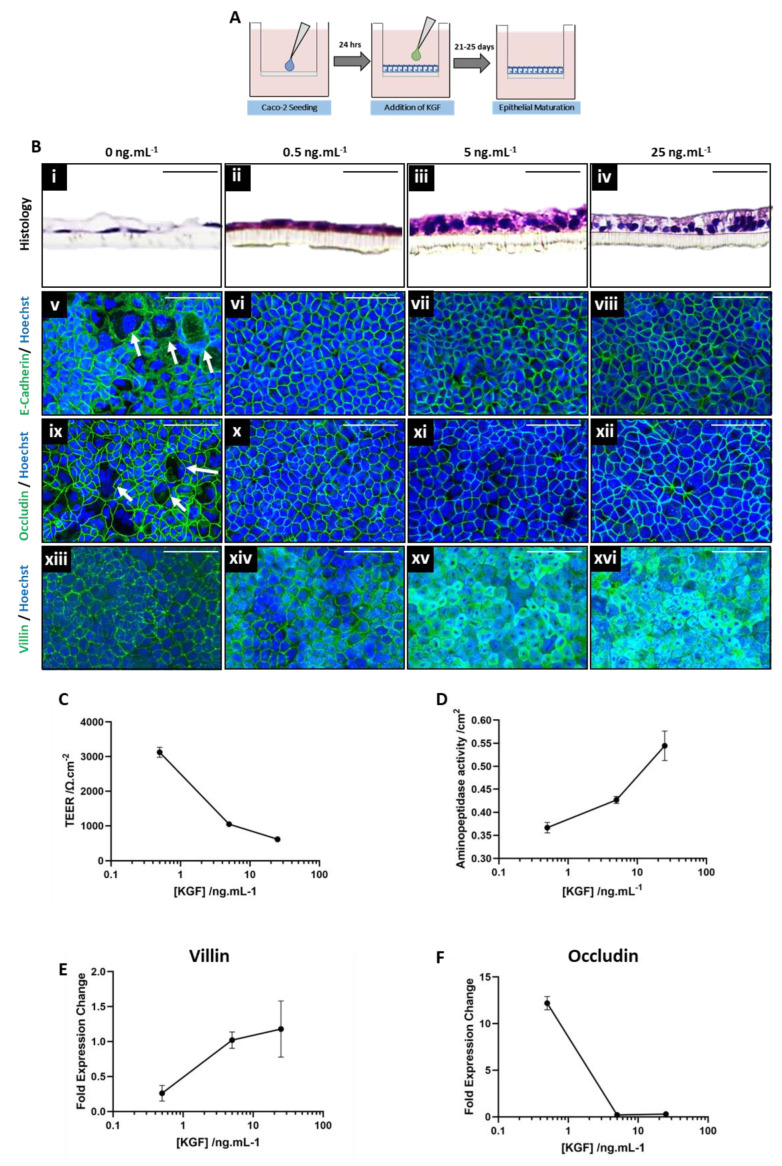
Kkeratinocyte growth factor impacts the structure and function of human bioengineered intestinal epithelial constructs. (**A**) A schematic representation of Caco-2 monolayers forming epithelial-only (EO) constructs and supplemented exogenously with a range of concentrations (0.5*–*25 ng.mL*^−^*^1^) of keratinocyte growth factor (KGF) after 24 h in culture and maintained for the entirety of the 25-day culture period. (**B**i*–*iv) H&E staining demonstrates the changing morphology of EO constructs with increasing concentration of KGF supplementation. Immunofluorescence analysis of junctional proteins (**B**ix*–*xii) E-cadherin (green) and (**B**ix*–*xii) occludin (green) reveal expression patterns in Caco-2 monolayers supplemented with KGF. Similarly, immunofluorescence detection of (**B**xiii-xvi) villin (green), a component of the microvilli that forms the brush border of enterocytes, increases with KGF concentration. All nuclei are stained blue by Hoechst. (**C**) Transepithelial electrical resistance (TEER), a functional assessment of epithelial barrier function, decreases with KGF supplementation (data represent mean ± SEM, n = 3). In contrast, (**D**) aminopeptidase, a digestive enzyme, activity increases with KGF concentration (data represent mean ± SEM, n = 3). qPCR analysis of genes responsible for expression of (**E**) villin and (**F**) occludin demonstrate increased and decreased expression with increasing KGF concentration, respectively (data represent mean ± SEM, n = 3). Scale bars: 50 μm.

**Figure 4 cells-13-01859-f004:**
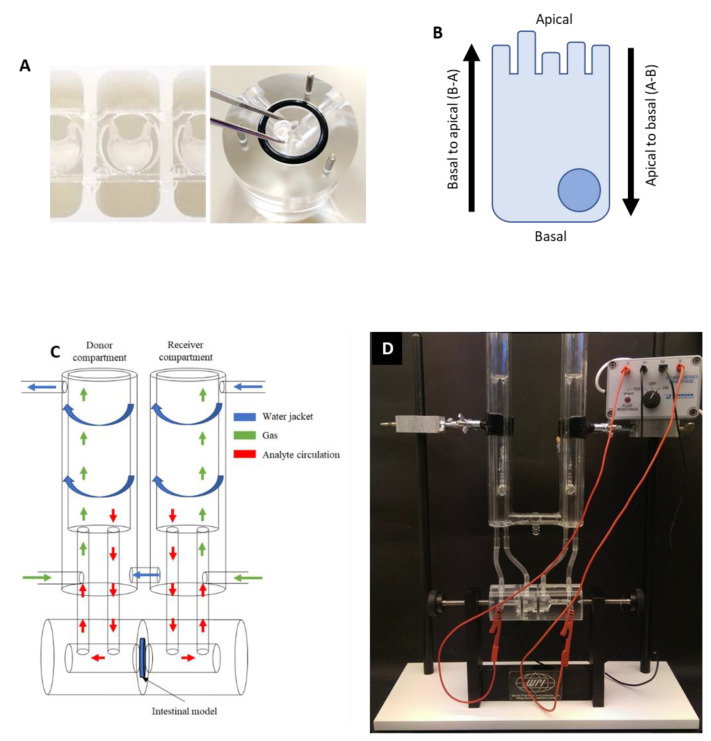
Ussing chamber apparatus used to assess the permeability of intestinal constructs in vitro. (**A**) A representative image of a 96-well Alvetex*^®^* Scaffold insert used to produce full-thickness (FT) intestinal constructs for physiological permeability screening (**left**), and the insertion of the insert into the chamber (**right**). (**B**) A schematic representation of the direction of transport as measured using the Ussing chamber, with transport either measured apical–basal (A-B) or basal–apical (B-A) depending on the orientation of the tissue construct. (**C**) A schematic representation of the Ussing system with a (**D**) supporting photograph. The system consists of two reservoirs, with which analytes and carbogen gas are circulated within a water jacket that maintains the system at a constant temperature of 37 °C. Bioengineered constructs (full-thickness or epithelial-only) reside at the interface between the reservoirs, the orientation of which dictates the direction of transport. Analytes are added or sampled from each reservoir, respectively.

**Figure 5 cells-13-01859-f005:**
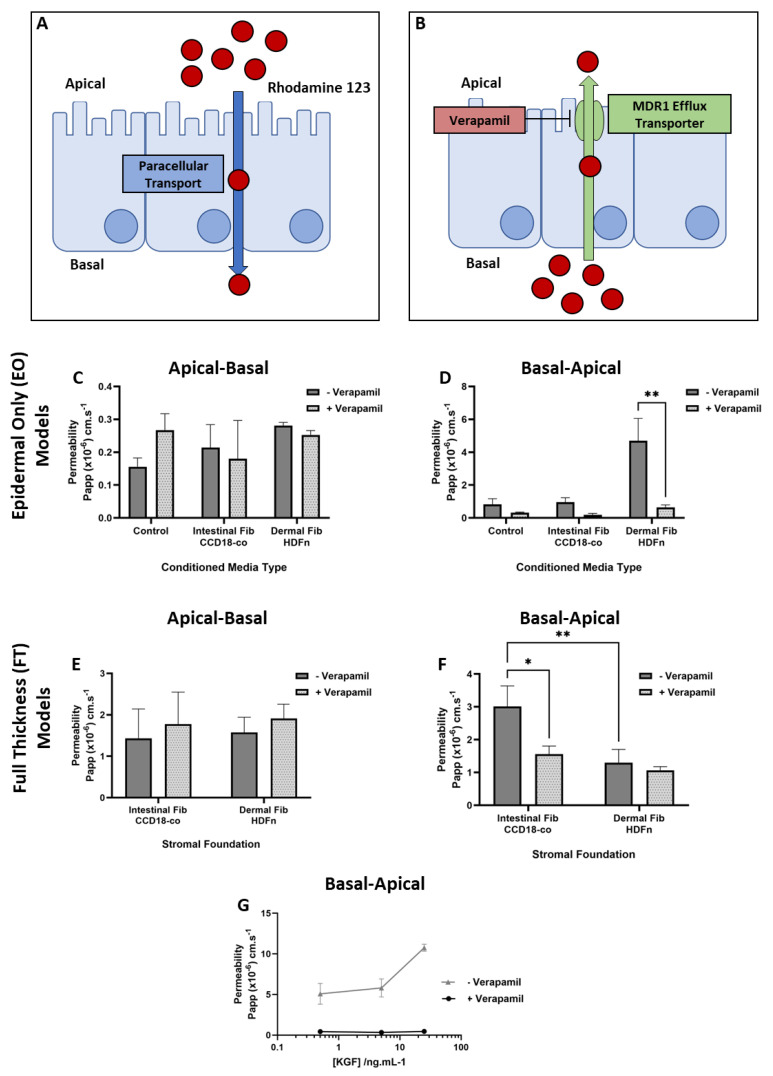
The Ussing chamber assessment of Rhodamine 123 transport in bioengineered intestinal constructs. (**A**) A schematic representation of apical–basal Rhodamine 123 transport mediated by paracellular diffusion and (**B**) basal–apical transport mediated by the efflux receptor: MDR1 and inhibited by verapamil. (**C**) The apparent permeability (Papp) of the Rhodamine 123 representative or apical–basal and (**D**) basal–apical transport in the presence or absence of verapamil across epithelial-only (EO) constructs cultured in standard medium (control), or medium conditioned by either intestinal or dermal fibroblast populations (data represent mean ± SEM, n = 3*–*5). A permeability assessment of full-thickness (FT) intestinal constructs built upon a stromal foundation of either intestinal or dermal fibroblasts, representative of (**E**) apical–basal and (**F**) basal–apical transport in the presence or absence of verapamil (data represent the mean ± SEM, n = 3*–*5). (**G**) The relationship between Rhodamine 123 permeability in the basal–apical direction in the presence or absence of verapamil of EO constructs exogenously treated with keratinocyte growth factor (KGF) at a range of concentrations (data represent the mean ± SEM, n = 3*–*5). * = *p* ≤ 0.1,** = *p* ≤ 0.01.

**Figure 6 cells-13-01859-f006:**
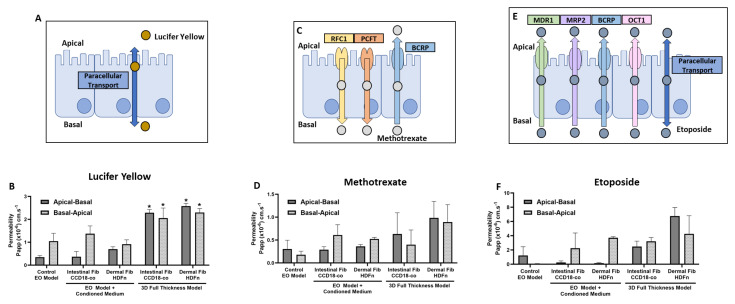
Permeability dynamics of bioengineered intestinal constructs in response to a number of standard compounds and drugs. (**A**) A schematic representation of lucifer yellow transport via paracellular diffusion. (**B**) The apparent permeability (Papp) of lucifer yellow in both epithelial-only (EO) constructs cultured in intestinal or dermal fibroblast conditioned mediums, and full-thickness constructs built upon an intestinal or dermal fibroblast stromal foundation (data represent mean ± SEM, n = 3–5). (**C**) A schematic representation of the anti-cancer drug, methotrexate, transport mediated by RFC1 and PCFT transporters in the apical–basal direction and the BCRP transporter in the basal–apical direction. (**D**) Papp of methotrexate in EO-conditioned medium constructs and FT co-culture constructs in both the apical–basal and basal–apical directions (data represent mean ± SEM, n = 3–5). (**E**) A schematic representation of cellular transport of the anti-tumor agent etoposide, transported in the basal–apical direction by a number of transporters, MDR1, MRP2, BCRP, and OCT1, and paracellular transport in the basal-apical direction. (**F**) Papp of etoposide in EO constructs cultured in fibroblast conditioned medium and FT co-culture constructs in both apical–basal and basal-apical directions (**F**) (data represent mean ± SEM, n = 3–5). * = *p* ≤ 0.1.

## Data Availability

Our supporting research data are published in the Durham University Research Data Repository. DOI: http://doi.org/10.15128/r26w924b89c.

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
