# Peer review of "Bioengineering the Human Intestinal Mucosa and the Importance of Stromal Support for Pharmacological Evaluation In Vitro"

_cells, 2024, doi:10.3390/cells13221859_

Round 1

Reviewer 1 Report

Comments and Suggestions for Authors

The authors provide an interesting, timely topic, paper. However, there are some aspects requiring consideration.

The Colonic cell line Caco-2 is widely used as a model of the intestinal barrier function. It has known limitations compared to in vivo conditions. Therefore several alternative in vitro models (e.g. Caco-2 subclones) were investigated.

You should implement a separate table highlighting the advantages and disadvantages comparing to your system.

Moreover, Caco-2 cells predominantly express human carboxylesterase 1 (hCE1), unlike the human intestine that predominantly expresses human carboxylesterase 2 (hCE2). This can lead to misestimating of the intestinal absorption (e.g. prodrugs).

Does your system offer advantages in this regard?

The substances to be examined must be dissolved in the Caco-2 model. For this reason, different formulations can only be tested to a limited extent.

Does your system offer advantages in this regard?

Author Response

We would like to express our gratitude to Reviewer 1 for taking the time to read and provide feedback on our manuscript. Changes made in accordance with Reviewer 1’s comments are highlighted in blue throughout the manuscript.

Comments for authors:

The authors provide an interesting, timely topic, paper. However, there are some aspects requiring consideration.

The Colonic cell line Caco-2 is widely used as a model of the intestinal barrier function. It has known limitations compared to in vivo conditions. Therefore several alternative in vitro models (e.g. Caco-2 subclones) were investigated.

You should implement a separate table highlighting the advantages and disadvantages comparing to your system.

We have included a table as requested to compare and contrast our models of intestinal absorption against others in the literature, please see table 1.

Moreover, Caco-2 cells predominantly express human carboxylesterase 1 (hCE1), unlike the human intestine that predominantly expresses human carboxylesterase 2 (hCE2). This can lead to misestimating of the intestinal absorption (e.g. prodrugs).

Does your system offer advantages in this regard?

We have not assessed expression of hCE1/2 or conducted gene expression analyses of any enzymes at this stage. However, we have added a sentence to the discussion to acknowledge this point:

“This includes structural hallmarks of improved epithelial morphology, increased permeability, cellular transporter expression and improved functionality, although ex-pression of metabolic enzymes was outside the scope of this study expression of which is known to be disparate expression in Caco2 monolayer cultures compared with in vivo.”

Lines 594-597

The substances to be examined must be dissolved in the Caco-2 model. For this reason, different formulations can only be tested to a limited extent.

Does your system offer advantages in this regard?

We have added drugs exogenously to the culture medium in either the apical or basal compartment, depending on directionality of testing. We believe this to be the most physiologically relevant method of drug presentation for oral drug testing, as when a candidate drug reaches the colon, it would be sufficiently broken down and presented to cells in solution. Our system is able to screen absorption of complex formulations, combinations of compounds or drugs in insolation depending on the study. Although outside the scope of this study, we would expect topical application of formulations to be possible within this system, as we have optimised topical delivery in our other bioengineered tissue systems. However, this means of treatment would better represent small intestine absorption, which is the reason it was not assessed in this study.

We have added the word formulation to our discussion to acknowledge this point:

“We have also demonstrated its assessment in a bespoke Ussing chamber, consistent with industrial standards, capable of high throughput drug, compound, and formulation screening, including a 96-well insert format.”

Line 685

Reviewer 2 Report

Comments and Suggestions for Authors

The present article by Matthew Freer et al, entitled "Bioengineering the human intestinal mucosa and the importance of stromal support for pharmacological evaluation in vitro", presents a humanised bioengineered intestinal construct, that recapitulates aspects of intestinal microanatomy. In brief, the improved histotypic characteristics reminiscent of human intestine, such as a reduction in transepithelial electrical resistance (TEER) and the formation of a robust basement membrane, which are contributed to in-part by a strong stromal foundation are described. Moreover, in this work, issues related to the the link between stromal-epithelial crosstalk, paracrine communication and the role of keratinocyte growth factor (KGF) as a soluble mediator, underpinning the the tissue-specific role of fibroblast subpopulations, are explored.

The manuscript is concisely written, well documented, cites an adequate number of relevant References and is of interest to the cognizant reader.

The methodologies used are appropriate and well described. A minor point that should be taken into account in the revised version of the manuscript is the lack of discussion about the infuence of the discrete stereoelectronic features of the model drugs used (methotrexate, etoposide) on their paracellular transport. These molecules due to their different functional groups interact, during their transport, via the formation of chemical  bonds of different nature (e.g. H-bonds, van der Waals, even London dispersion forces). 

Author Response

We would like to express our gratitude to Reviewer 2 for taking the time to read and provide feedback on our manuscript.

Comments for author:

The present article by Matthew Freer et al, entitled "Bioengineering the human intestinal mucosa and the importance of stromal support for pharmacological evaluation in vitro", presents a humanised bioengineered intestinal construct, that recapitulates aspects of intestinal microanatomy. In brief, the improved histotypic characteristics reminiscent of human intestine, such as a reduction in transepithelial electrical resistance (TEER) and the formation of a robust basement membrane, which are contributed to in-part by a strong stromal foundation are described. Moreover, in this work, issues related to the the link between stromal-epithelial crosstalk, paracrine communication and the role of keratinocyte growth factor (KGF) as a soluble mediator, underpinning the the tissue-specific role of fibroblast subpopulations, are explored.

The manuscript is concisely written, well documented, cites an adequate number of relevant References and is of interest to the cognizant reader.

The methodologies used are appropriate and well described. A minor point that should be taken into account in the revised version of the manuscript is the lack of discussion about the infuence of the discrete stereoelectronic features of the model drugs used (methotrexate, etoposide) on their paracellular transport. These molecules due to their different functional groups interact, during their transport, via the formation of chemical  bonds of different nature (e.g. H-bonds, van der Waals, even London dispersion forces). 

The drugs and molecules tested in this study, are all widely commercially available and well characterised in terms of intestinal absorption. They provide proof-of-concept evidence, demonstrating the application range of the constructs developed in this manuscript, as a means to test bi-directional permeability in our systems, the specific biochemical interactions at a molecular level are outside the scope of this study.

Reviewer 3 Report

Comments and Suggestions for Authors

The manuscript presents the development and characterization of an in vitro intestinal construct, that could overcome some limitation of the actual model used for pharmacokinetic testing, the Caco-2 epithelium approach, and that might be used for high throughput compound screenings. In my opinion, the manuscript covers an interesting topic, the research is well conducted, supported by a lot of experimental work, the results are clearly presented,  and the paper is well written. The paper can be accepted in its current form.

Author Response

We would like to express our thanks to Reviewer 3 for endorsing our article for publication.

Reviewer 4 Report

Comments and Suggestions for Authors

Through a combination of tissue specific fibroblasts and a 3D geometry allowing cell-cell and cell-matrix interactions, authors have demonstrated a FT intestinal construct that better recapitulates intestinal permeability compared with the existing Caco-2 mono-culture approach.

In fact, this paper explains the limitations of the traditional Caco-2 monolayer but could provide a more detailed comparison with other advanced in vitro models. A comparative analysis is strongly recommended.

While the scope is clear, the rationale could be strengthened by directly contrasting how current models fail to predict in vivo drug absorption for different classes of compounds (even considering the SM).

For instance, more emphasis on specific cases of failure in drug prediction could better justify the development of an advanced model.

It lacks sufficient detail on how the proposed model performs relative to other cutting-edge models (summary in a table).

This approach has another limitations such as artificially high TEER and poor paracellular permeability. Authors should explain these isssues.

Besides, this paper eliminates the use of animal that are ubiquitous amongst other published intestinal constructs.

 Authors stated that although paracrine crosstalk between compartments is an important factor, geometry and direct cell-cell contact is also a critical consideration when bioengineering epithelial tissue.

Author Response

We would like to express our gratitude to Reviewer 4 for taking the time to read and provide feedback on our manuscript. Changes made in accordance with Reviewer 4’s comments are highlighted in pink throughout the manuscript.

Comments for authors:

Through a combination of tissue specific fibroblasts and a 3D geometry allowing cell-cell and cell-matrix interactions, authors have demonstrated a FT intestinal construct that better recapitulates intestinal permeability compared with the existing Caco-2 mono-culture approach.

In fact, this paper explains the limitations of the traditional Caco-2 monolayer but could provide a more detailed comparison with other advanced in vitro models. A comparative analysis is strongly recommended.

While the scope is clear, the rationale could be strengthened by directly contrasting how current models fail to predict in vivo drug absorption for different classes of compounds (even considering the SM).

For instance, more emphasis on specific cases of failure in drug prediction could better justify the development of an advanced model.

We have included the following sentence to the introduction, to provide specific examples of drugs absorption in Caco2 monolayers that do not accurately reflect that of the in vivo state:

“Examples of which include hydrophilic drugs where low Caco-2 permeability in vitro does not correlate with high absorption of these compounds in vivo such as: amoxicillin, cefadroxil, cephalexin, loracarbef, pregabalin, and zidovudine [14].”

Lines 48-51

It lacks sufficient detail on how the proposed model performs relative to other cutting-edge models (summary in a table).

In order to compare and contrast the model detailed in this manuscript, with those described in the literature, we have included a table of comparison (Table 1). This table details all the above points made by Reviewer 4. A comparison of approaches described in the literature, and how the models we describe in this manuscript perform compared to them.

This approach has another limitations such as artificially high TEER and poor paracellular permeability. Authors should explain these isssues.

We added the following lines to the discussion to explain this point:

“… both of which reduce the absorptive capacity of the construct and therefore its physiological relevance compared with native tissue”

Lines 591-592

Besides, this paper eliminates the use of animal that are ubiquitous amongst other published intestinal constructs.

 Authors stated that although paracrine crosstalk between compartments is an important factor, geometry and direct cell-cell contact is also a critical consideration when bioengineering epithelial tissue.